# Identification of *Leishmania infantum* Epidemiology, Drug Resistance and Pathogenicity Biomarkers with Nanopore Sequencing

**DOI:** 10.3390/microorganisms10112256

**Published:** 2022-11-14

**Authors:** Joan Martí-Carreras, Marina Carrasco, Marcel Gómez-Ponce, Marc Noguera-Julián, Roser Fisa, Cristina Riera, Maria Magdalena Alcover, Xavier Roura, Lluís Ferrer, Olga Francino

**Affiliations:** 1Nano1Health S.L. (N1H), Edifici EUREKA, Parc de Recerca UAB, Bellaterra, 08193 Barcelona, Spain; 2Laboratori de Parasitologia, Departament de Biologia Sanitat i Mediambient, Facultat de Farmàcia I Ciències de l’Alimentació, Universitat de Barcelona, Av. Joan XXIII 27-31, 08028 Barcelona, Spain; 3Hospital Clínic Veterinari, Universitat Autònoma de Barcelona, Bellaterra, 08193 Barcelona, Spain; 4Departament de Medicina i Cirurgia Animals, Universitat Autònoma de Barcelona, Bellaterra, 08193 Barcelona, Spain

**Keywords:** *Leishmania infantum*, leishmaniosis, drug resistance, treatment, nanopore sequencing, copy number variation, aneuploidy, maxicircle, LeishGenApp

## Abstract

The emergence of drug-resistant strains of the parasite *Leishmania infantum* infecting dogs and humans represents an increasing threat. *L. infantum* genomes are complex and unstable with extensive structural variations, ranging from aneuploidies to multiple copy number variations (CNVs). These CNVs have recently been validated as biomarkers of *Leishmania* concerning virulence, tissue tropism, and drug resistance. As a proof-of-concept to develop a novel diagnosis platform (LeishGenApp), four *L. infantum* samples from humans and dogs were nanopore sequenced. Samples were epidemiologically typed within the Mediterranean *L. infantum* group, identifying members of the *JCP5* and *non-JCP5* subgroups, using the conserved region (CR) of the maxicircle kinetoplast. Aneuploidies were frequent and heterogenous between samples, yet only chromosome 31 tetrasomy was common between all the samples. A high frequency of aneuploidies was observed for samples with long passage history (MHOM/TN/80/IPT-1), whereas fewer were detected for samples maintained in vivo (MCRI/ES/2006/CATB033). Twenty-two genes were studied to generate a genetic pharmacoresistance profile against miltefosine, allopurinol, trivalent antimonials, amphotericin, and paromomycin. MHOM/TN/80/IPT-1 and MCRI/ES/2006/CATB033 displayed a genetic profile with potential resistance against miltefosine and allopurinol. Meanwhile, MHOM/ES/2016/CATB101 and LCAN/ES/2020/CATB102 were identified as potentially resistant against paromomycin. All four samples displayed a genetic profile for resistance against trivalent antimonials. Overall, this proof-of-concept revealed the potential of nanopore sequencing and LeishGenApp for the determination of epidemiological, drug resistance, and pathogenicity biomarkers in *L. infantum*.

## 1. Introduction

*Leishmania infantum* is a parasitic protozoan of the order *Trypanosomatida* (family *Trypanosomatidae*), known to infect and cause severe disease in dogs and other mammals which can act, in turn, as reservoirs for zoonotic transmission to humans (further developed in Hong et al., 2020) [1]. *L. infantum* is transmitted as an extracellular parasite (promastigote) to a mammalian host by female infected phlebotomine sand flies. Afterward, the parasites proliferate as obligate intracellular parasites (amastigotes) in phagocytic cells and spread to different organs, causing leishmaniosis. The infection in the mammalian host can occur subclinically or with clinical signs of varying severity. In humans, *L. infantum* infection can cause cutaneous lesions (cutaneous leishmaniosis or CL) and systemic disease (visceral leishmaniosis or VL). In dogs, the infection causes a severe systemic disease, with cutaneous lesions, lymphadenopathy, weight loss, and chronic kidney disease, among other clinical signs. Dogs are the major domestic reservoir of *L. infantum*, but wild mammals such as the red fox, black rats, and the Iberian hare can also be involved in the transmission cycle [1,2,3].

Several drugs are available to treat leishmaniosis, both in humans and dogs, albeit their effectiveness may vary and can be considered toxic. Trivalent antimonials and miltefosine, although potentially hepatotoxic and nephrotoxic (reviewed in van Griensven and Diro, 2019 [4]), are the first-choice treatment in dogs and are also used to treat human patients. As in many infections, diversity and selection have given rise to drug resistance and treatment failure. Given the limited therapeutic arsenal, the increasing emergence of resistant *L. infantum* strains is a serious global health problem.

The genome of *Leishmania infantum* is diploid with an approximate size of 32 Mb, organized in 36 chromosomes (plus maxicircles and minicircles, which form the kinetoplast network). Genome instability is characteristic of *L. infantum* and of all the *Leishmania* genus, and it is responsible for unique genomic characteristics such as (i) mosaic aneuploidy [5] and (ii) transcription regulation by gene copy number variation (CNV) [6]. The genome instability is triggered by heterologous recombination of direct and indirect repeats spread along its genome [7]. This stochastic rearrangement (“genome *sizing and shuffling*”) may introduce local or chromosome wide CNVs, reaching its characteristic mosaic subpopulations and eliciting a fast adaptation to changing environments [8]. Thus, studies that identify mutations or genetic markers as well as the underlying mechanic for drug resistance are essential for determining specific genetic variations involved in drug resistance mechanisms; reviewed in Ponte-Sucre et al., 2017 [9] as well as for parasite population genetics and epidemiology. Some genetic variations have been associated with resistance to drugs: an increase in copy number of genes in the H locus (chromosome 23) has been associated with trivalent antimonial resistance, and a reduction in *METK* (LinJ.30.3560) gene copies has been associated with increased allopurinol resistance [10]. Other markers, such as the coding regions of the conserved region (CR) of the maxicircle, have recently been reported as suitable for *Leishmania* taxonomy for phylogenetic analysis, including subspecies divisions [11,12].

Rapid aneuploidy and CNV turnover (“genome *sizing* and *shuffling*”) serve as a substrate for variability that, combined with selection, triggers a fast environment adaptation and enhances parasite evolvability under stress (i.e., culture, extra- and intracellular life cycle, drug resistance, etc.) [13]. Massive short-read sequencing has proven valuable for single nucleotide polymorphism (SNP) discovery but lacks power for more complex structural variants [14]. Long reads from single-molecule sequencing span repetitive regions of the genome, facilitating subsequent assembly of genomes and offering an added advantage in quantitatively identifying and assessing structural variations and gene amplification events [14]. Furthermore, long-read technologies allow sequencing without the need for previous amplification, so it is possible to directly compare the coverage between different regions for biomarker panels or regions with very divergent GC content [15].

In this pilot study, we present the development of a genomic analysis platform (LeishGenApp) that allows sequencing and bioinformatics analyses to identify pharmacoresistance, pathogenicity, and population genetic biomarkers for *L. infantum*. In this proof-of-concept, the validity of direct uncorrected nanopore reads is defended for identifying and typing *L. infantum* samples and detecting genomic aneuploidies and gene copy number variations, (un)known markers for pathogenicity, and pharmacoresistance. Further development will be required to transform this proof-of-concept into an agile and straightforward genomic platform for use in clinical practice, and as an aid in conducting epidemiological studies or on treatment and prognosis of leishmaniosis.

## 2. Materials and Methods

### 2.1. Samples

Four *Leishmania infantum* samples were acquired to test as a proof-of-concept: (i) ATCC Control DNA from reference *L. infantum* strain Nicolle (ATCC 50134D, MHOM/TN/80/IPT-1) and (ii) three promastigote cultures received from Facultat de Farmàcia (Universitat de Barcelona, Barcelona, Spain) collections. The cultured samples were propagated in Schneider medium (S0146-500ML, Sigma-Aldrich, St. Louis, MO, USA) supplemented with 20% fetal bovine serum (FBS), 1% human urine, and 25 mg/mL gentamicin. Cultured samples correspond to MHOM/ES/2016/CATB101, LCAN/ES/2020/CATB102, and MCRI/ES/2006/CATB033 (isolated from dog’s skin exudate and passaged in *Cricetus aureus*) from the north-western Mediterranean region (Table 1).

### 2.2. DNA Extraction and Sequencing

DNA was extracted from the three cultures using the ZymoBIOMICS DNA Miniprep Kit (D4300; Zymo Research Corporation; Los Angeles, CA, USA), following the manufacturer’s recommendations. Before library preparation, extracted DNA was quantified using Qubit dsDNA BR Assay Kit (Fisher Scientific S.L; Madrid, Spain), and quality was assessed by absorbance using a NanoDrop 2000 spectrophotometer (ThermoFisher Scientific S.L; Waltham, MA, USA). The sequencing libraries were prepared using the Rapid Barcoding Sequencing Kit (SQKRBK004) from Oxford Nanopore Technologies (ONT, Oxford, UK), following the manufacturer’s recommendations. Rapid sequencing barcodes were added to the tagged ends to analyze more than one sample in a single run (barcoding). As a final step, 12 μL of each library was loaded onto a flow cell for sequencing using the Mk1c (MinION, ONT, Oxford, UK) and run for 48 h.

### 2.3. Bioinformatic Analysis

#### 2.3.1. Identification, Typing, and Phylogeny

Guppy (v6.1.2; ‘SUP’ model) was used for basecalling and demultiplexing all runs. Species identification was carried out using *blastn* (part of BLAST+ v2.9.0 [16] with direct uncorrected reads against a local database of *Trypanosomatidae,* downloaded from NCBI-NIH records by taxonomical ID (nucleotide database with txid5654). Typing was conducted by reconstruction of the conserved region of the maxicircle kinetoplast by mapping the corresponding reads with minimap2 v2.17-r941 [17] against the *L. infantum* JPCM5 v2/2018 reference (http://leish-esp.cbm.uam.es/l_infantum_downloads.html, accessed on 1 April 2022) [18] and deriving the consensus sequencing using SAMTools v1.10 [19] and BCFtools v1.10.2 [20]. Phylogenetic data from previous studies were collected from the Leish-ESP site (http://leish-esp.cbm.uam.es/l_infantum_downloads.html, accessed on 1 April 2022) [21]. *L. infantum* maxicircle phylogeny was constructed by multiple sequence alignment concatenation of conserved genes sequences 12S rRNA, 9S rRNA, ND8, ND9, MURF5, ND7, CO3, CYb, ATPase 6, ND2, G3, ND1, CO2, MURF2, CO1, G4, ND4, G5 (ND3), RPS12, and ND5 using MAFFT v7.490 [22]. Phylogeny was reconstructed using the maximum likelihood method applying the Tamura–Nei 1993 model [23] with 1000 bootstraps, selecting the topology with the best log-likelihood with IQ-Tree2 v2.2.0 [24].

#### 2.3.2. Aneuploidy and Gene Copy Number Variation Analysis

Aneuploidy and gene copy number variation were determined by calculating the log_2_ change of observed vs. expected copies (2N) over a sliding window of 100 kbp and 1 kbp, respectively. Thus, the copy ratio of 0 is 2N, 0.5 is 3N, 1 is 4N, or −1 is N. Both analyses were conducted using the LeishGenApp analysis platform (Nano1Health S.L., Bellaterra, Spain). Briefly, direct uncorrected nanopore reads were mapped against a reference using LRA v1.3.4 [25] and the read alignments were processed by SAMTools v1.10 [19]. The coverage in read alignments was screened with multiple sliding windows (as previously described) with CNVkit v0.9.8 [26].

Chromosome variation was significant by a threshold of ±0.2, as recommended elsewhere [26]. For instance, all the copy ratio values between −0.2 and 0.2 were determined as diploid (2N). Locally, nine genomic regions harboring 22 genes previously described as related to pathogenicity or drug resistance in *L infantum* have been used as a proof-of-concept for CNV characterization (Table 2). *L. infantum* JPCM5 v2/2018 genome assembly and annotation were used as reference for CNV (http://leish-esp.cbm.uam.es/l_infantum_downloads.html, accessed during 1 April 2022) [18]. Gene identification is conserved with previous assemblies, as described elsewhere [18]. A more stringent threshold, ±0.25, was assigned to identify variation in gene copy number than for aneuploidy detection, accounting for a greater effect of population mosaicism of smaller sliding windows. If multiple copy numbers are reported due to mosaicism, the most extreme copy number is reported by convention. Gene copy number was calculated from log_2_ for each of the 22 genes. CNVs were expressed as an addition to the expected number of chromosomes or gene copies (2N).

### 2.4. Data Availability

Raw sequencing data for this experiment can be accessed at NCBI-NIH SRA SRR21601459—SRR21601462, BioSamples SAMN30884654—SAMN30884657, under BioProject SUB12055562.

## 3. Results

### 3.1. Identification, Typing, and Phylogeny

Each culture (MHOM/TN/80/IPT-1, MHOM/ES/2016/CATB101, LCAN/ES/2020/CATB102, and MCRI/ES/2006/CATB033) was successfully identified as belonging to the *Leishmania infantum* species by aligning direct uncorrected nanopore reads against a local database of *Trypanosomatidae* sequences with *blastn* (best hit with e-value < 1 × 10^−8^). For each sample, 94% (99% *L. donovani*–*L. infantum* complex), 94% (99% *L. donovani*–*L. infantum* complex), 94% (99% *L. donovani*–*L. infantum* complex), and 96% (99% *L. donovani*–*L. infantum* complex) of total reads were assigned to *L. infantum*, respectively. The remaining 1% was assigned to other *Leishmania spp.* complexes (*L. major* or *L. mexicana*) or unannotated.

Each CR sequence could be successfully reconstructed with a variety of mean coverages: 161X for MHOM/TN/80/IPT-1, 15X for MHOM/ES/2016/CATB101, 8X for LCAN/ES/2020/CATB102, and 3X for MCRI/ES/2006/CATB033. Sequence alignment and a maximum likelihood (ML) tree of the CR of the maxicircle with the *Leishmania–Trypanosoma* phylogeny procured by Solana et al., 2022 placed them within the phylogenetic cluster of *L. infantum* (Appendix A). Moreover, as shown in Figure 1, when placing the four sequences (marked in red) in an *L. donovani*–*L. infantum* phylogeny, the novel sequences still cluster within *L. infantum* but are divided between the so-called *JPC5* subgroup (marked in dark green) and *non-JPC5* subgroup (marked in blue) clusters. MHOM/ES/2016/CATB101 is the only sequence that clusters in the alternative *non-JPC5* subgroup Spanish cluster. Thus, shallow coverage (<10X) does not interfere with sequence reconstruction nor with its phylogenetic placement.

#### 3.1.1. Aneuploidy Analysis

Regarding the aneuploidy analysis, the chromosomal dotation of the samples was heterogeneous (Figure 2). Chromosome 31 was tetrasomic in all the studied samples. Total or mosaic trisomy was observed for chromosomes 5, 9, 11, 20, 21, 24, 25, 26, and 29 for sample MHOM/TN/80/IPT-1; chromosomes 1, 5, 9, 11, 23, 26, and 35 for sample MHOM/ES/2016/CATB101; and chromosomes 33 and 35 for LCAN/ES/2020/CATB102. Changes involving chromosome gain (Figure 2, green, turquoise, and violet) were more common among all samples than those involving chromosome loss (red). Sample MHOM/TN/80/IPT-1 harbors the highest number of chromosomal variations (nine trisomies, four tetrasomies, and one hexasomy). Other anomalies identified were hexasomy for chromosome 23 in sample MHOM/TN/80/IPT-1, and partial monosomy (mosaic) for chromosome 13 in sample MCRI/ES/2006/CATB033. This finding supports the presence of intrastrain mosaicism and interstrain chromosomal diversity.

#### 3.1.2. CNV for Drug Resistance and Pathogenicity Biomarkers in *L. infantum*

Miltefosine, allopurinol, trivalent antimonials, amphotericin, and paromomycin are the most used drugs used to treat leishmaniosis. As depicted in Table 2, CNVs of these 22 genes may act as potential biomarkers for pharmacoresistance in *L. infantum*. All 22 genes were detected with a coverage > 5X. CNVs were detected in all the samples: 13 genes with CNV in LCAN/ES/2020/CATB102, 16 genes with CNV in MCRI/ES/2006/CATB033, 18 genes with CNV in MHOM/TN/80/IPT-1, and 19 genes with CNV in MHOM/ES/2016/CATB101 (Table 3). Hence, nine, six, four, and three genes remain copy-neutral (CN 0) for each sample. Variation was observed for all the genes in at least one sample. Overall, the variation in copy number compared to the diploid dotation ranged from +7 copies for the LinJ.23.0280 gene (*YIP1*) in MHOM/TN/80/IPT-1 to −1 for the LinJ.36.6760 gene (*LMPK*) in MCRI/ES/2006/CATB033.

The miltefosine transporter and associated genes are in chromosomes 13 (LinJ.13.1590, *LdMT*; LinJ13.1600, *hypothetical protein* gene) and 32 (LinJ.32.1040, *IdRos3*) [32], where local deletions (CN −1) were found in three strains (MHOM/TN/80/IPT-1, MHOM/ES/2016/CATB101, and MCRI/ES/2006/CATB033). Regarding the miltefosine sensitivity locus (*MSL*; LinJ.31.2370 to LinJ.31.2400), located in chromosome 31 (tetrasomic) [27], local expansions were detected at the population level in all four samples. In addition to the duplication of the entire diploid chromosome dotation (CN +2), additional local expansions of this entire locus were detected in sample MHOM/TN/80/IPT-1 (CN +4; +4; +4, +5, and +3 in LinJ.31.2370 to LinJ.31.2400, respectively). Considering the tetrasomic dotation of chromosome 31, partial deletions (CN +1, +2) were observed in samples MHOM/ES/2016/CATB101 and LCAN/ES/2020/CATB102.

Allopurinol pharmacoresistance is linked to the *METK* locus located in chromosome 30, which contains four genes (LinJ.30.3550, *Lorien protein* gene; LinJ.30.3560, *METK1*; LinJ.30.3570, *Lorien protein* gene; LinJ.30.3580, *METK2*) [10]. Samples MHOM/ES/2016/CATB101 and MCRI/ES/2006/CATB033 presented local expansion (CN +1) in all genes in the *METK* locus, while MHOM/TN/80/IPT-1 and LCAN/ES/2020/CATB102 followed a different trend with partial deletions in LinJ.30.3560 and LinJ.30.3570.

The H locus, *AQP1* gene, *MAPK1* gene, and *SMT* gene are trivalent antimonial resistance biomarkers [30,31,33]. The H locus harbors four genes in chromosome 23 (LinJ.23.0280, *YIP1*; LinJ.23.0290, *MRPA*; LinJ.23.0300, LinJ.23.0300; LinJ.23.0310, *PTR1*). Samples MHOM/ES/2016/CATB101, LCAN/ES/2020/CATB102, and MCRI/ES/2006/CATB033 showed CNV and mosaicism (CN +1). Remarkably, MHOM/TN/80/IPT-1 had the greatest increase that ranged from +4 in *PTR1* to +7 in *YIP1*. These local expansions could be related to the ploidy of chromosome 23 in the four samples (hexasomy in MHOM/TN/80/IPT-1, partial trisomy in LCAN/ES/2020/CATB102, trisomy in MHOM/ES/2016/CATB101, and disomy in MCRI/ES/2006/CATB033). The *AQP1* gene (LinJ.31.0030) is in chromosome 31 (tetrasomic). Considering the duplication of the complete chromosome dotation, an additional gene expansion was observed in sample MHOM/TN/80/IPT-1 (CN +5). Total gene (CN +1) deletion was observed in sample MHOM/ES/2016/CATB101, and partial (CN +1, +2 and +2, +3) deletions were observed in MCRI/ES/2006/CATB033 and LCAN/ES/2020/CATB102, respectively. The *MAPK1* gene (LinJ.36.6760) and *SMT* gene (LinJ.36.2510) copy number ranged from −1 to +1 CN.

The paromomycin resistance locus entails two genes in chromosome 27 (LinJ.27.1940, *D-LDH*; LinJ.27.1950, *B-CAT*) [30]. While the *D-LDH* had two copies (CN 0) in all the samples except in LCAN/ES/2020/CATB102 (CN 0, +1), the *B-CAT* gene copy number presented certain variability among the samples. MHOM/TN/80/IPT-1 had CN −1 compared to the reference, MHOM/ES/2016/CATB101, MCRI/ES/2006/CATB033 had CN 0, and LCAN/ES/2020/CATB102 had CN +1.

The LACK antigen is a protein encoded by *Lack1* (LinJ.28.2940) and *Lack2* (LinJ.28.2970) genes, and they are related to *Leishmania* pathogenicity [29]. Both are located at chromosome 28. Mosaicism of both genes (CN 0, +1) was observed in MHOM/TN/80/IPT-1 and MHOM/ES/2016/CATB101; *Lack2* was present as a mosaic (CN 0, +1) in LCAN/ES/2020/CATB102. MCRI/ES/2006/CATB033 had no CNV in this locus.

## 4. Discussion

### 4.1. Identification, Typing, and Phylogeny

All four samples (MHOM/TN/80/IPT-1, MHOM/ES/2016/CATB101, LCAN/ES/2020/CATB102, and MCRI/ES/2006/CATB033) were successfully identified as *Leishmania infantum*. Direct uncorrected nanopore reads proved to be of sufficient quality and length to successfully assign at least 94% (max. 96%) of the total reads to *L. infantum* with e-value < 1× 10^−8^, further validating the technology as a feasible approach for pathogen identification. The remaining reads were assigned to members of the *L. donovani*–*L. infantum* complex (5%) or were unassigned or misassigned to other *Trypanosomatidae* members (1%). Differentiation between *L. infantum* and *L. donovani* can be difficult due to their high nucleotide identity (i.e., chromosome 36 is 99.16% similar between references *L. donovani* BPK282A1 and *L. infantum* JPCM5); thus, it is not striking that a small percentage of reads (5%) is assigned as *L. donovani*. The remaining 1% consists of a mixture of assignments to other *Trypanosomatidae*, to high error reads or sequences belonging to the maxicircle and minicircle kinetoplasts, for which there is no complete reference for *L. infantum* yet [12,21].

In addition to species identification by random sequence fragments, maxicircle kinetoplast sequences from *Trypanosomatidae* have been previously described as a suitable molecular marker for species [11,21] and strain [12] phylogenies, akin to mitochondrial sequences from the kingdoms *Plantae*, *Fungi,* or *Animalia*. As shown in Appendix A, reconstruction of the *L. infantum* conserved region (CR) of the maxicircle is possible through guided consensus assembly with nanopore sequencing reads, with as little coverage as with a mean of 3X. Such new CR sequences are similar in quality to those previously published from *L. infantum*, as they cluster together in a *Leishmania–Trypanosoma* phylogeny. Moreover, as shown in Figure 1, group typing is possible within the *L. donovani*–*L. infantum* complex with these reconstructed sequences. The CRs of MHOM/TN/80/IPT-1, MCRI/ES/2006/CATB033, and LCAN/ES/2020/CATB102 isolates correspond to the classic JPC5-like group, which is very prevalent in the Iberian Peninsula and present in other Mediterranean areas. Interestingly, isolate MHOM/ES/2016/CATB101 is placed within the *non-JPC5* cluster as it possesses the same 17 SNPs described in Solana et al., 2022 [12], showing that the non-JPC5 *L. infantum* group was circulating at least two years prior than previously estimated with isolates MHOM/ES/2018/LLM-2404, MHOM/ES/2018/LLM-2406, MHOM/ES/2018/LLM-2408, MHOM/ES/2018/LLM-2409, and MHOM/ES/2018/LLM-2410, dated from 2018. Beyond establishing an earlier timeline, MHOM/ES/2016/CATB101 establishes the presence of a non-JPC5 *L. infantum* group out of the Iberian Peninsula (Mallorca, Balearic Islands).

### 4.2. Detection of Aneuploidy and Gene Copy Number Variation

*Leishmania* parasites mostly rely on aneuploidy and DNA CNVs to regulate the expression of stress response genes to, e.g., temperature, acidity, or drugs [7,34,35]. As presented in the Results section, direct uncorrected nanopore reads have been suitable to detect aneuploidy, including chromosome mosaicism, and previously described CNV related to genetic drug resistance biomarkers.

Notably, local CNVs of targeted chromosome regions were more common across chromosomes in all four samples than aneuploidy. These genome plasticity strategies are a good solution for transcript regulation for an organism that lacks promoter-dependent regulation [6]. However, with this approach, it is not possible to discern whether this local gene amplification is intrachromosomal, maintained as extrachromosomal (linear or circular) molecules [34], or an expansion of an entire chromosome. Therefore, further studies to conclude its physical conformation should be conducted.

Remarkably, when comparing the ploidy of the four samples, MCRI/ES/2006/CATB033 has the smallest number of aneuploidies. Furthermore, it is the only sample where gene loss was quantified at the chromosomal level (sample is mosaic for a monoploidy of chromosome 13). It is noteworthy that MCRI/ES/2006/CATB033, despite being isolated from a dog, was propagated in hamsters (*Cricetus aureus*), in vivo, in contrast with the other three samples, which were only maintained as in vitro cultured promastigotes. MHOM/TN/80/IPT-1 had the largest number of aneuploidies (n = 14), probably linked to its long propagation and culture history, as it was isolated in 1980. Such long culture history may have adapted that strain to culturing conditions, as it is known that parasite isolation and subsequent in vitro parasite maintenance are strong drivers for chromosome and gene copy number variation [36,37]. Moreover, according to Domagalska et al., aneuploidy was much lower in amastigotes (intracellular stage of the parasite) than in cultivated promastigotes (extracellular) [37]. Our results are consistent with previous studies, given that the closest diploid karyotype was found in the in vivo MCRI/ES/2006/CATB033. Otherwise, MHOM/TN/80/IPT-1 showed the highest number of aneuploidies due to the in vitro effect on ploidy variation [37]. Considering changes in chromosome copy number as a highly common feature during experimental selection, the results support the importance of minimizing culture laboratory passaging or direct clinical samples when studying aneuploidy and CNV as genomic biomarkers.

Local detection of CNVs in the 22 genes studied revealed possible pharmacoresistance in our sample collection. A deletion in *LdMT* and/or *ldRos3* (CN −1) is related with a 2- and 1.6-fold decrease in miltefosine sensitivity, respectively [32]. According to this genetic biomarker, samples MHOM/TN/80/IPT-1 and MCRI/ES/2006/CATB033 have the genetic potential to be pharmacoresistant to miltefosine. Additionally, the same samples have the genetic potential to have allopurinol pharmacoresistance as a deletion (CN −1) in the *METK1* gene was quantified [10]. Regarding biomarkers for trivalent antimonial resistance, all four strains showed genetic potential for pharmacoresistance since an additional copy (CN +1) of the *MRPA* gene in the H locus is related to an increase in resistance [28]. *MRPA* CNVs ranged from +1 in MCRI/ES/2006/CATB033 to CN +5 in MHOM/TN/80/IPT-1. Finally, an extra copy of *D-LDH* and *B-CAT* genes is linked with a 4.87 and 4.08-fold increase in paromomycin resistance, respectively [30]. Potential genetic pharmacoresistance could be detected in MHOM/ES/2016/CATB101 and in LCAN/ES/2020/CATB102 since mosaic expansions (CN 0, +1) were found in *B-CAT* in both strains and *D-LDH* in the latter. A larger cohort of samples and phenotypic data (resistance) are required to validate the clinical relevance of these genetic pharmacoresistance profiles. A current limitation of this methodology is the elucidation of the physical conformation of aneuploidies and gene CNs (chromosomal or extrachromosomal). Despite not being relevant for possible genetic pharmacoresistance or virulence, this difference is relevant for the transmission of virulence, as described for *Leishmania spp.* Small extrachromosomal circles with virulence genes can be readily transmitted through vesicle transport to neighboring parasites [34]. Thus, an exhaustive analysis should be carried out in further studies to overcome this limitation.

## 5. Conclusions

Direct uncorrected nanopore reads were obtained from four samples of *Leishmania infantum* (MHOM/TN/80/IPT-1, MHOM/ES/2016/CATB101, LCAN/ES/2020/CATB102, and MCRI/ES/2006/CATB033). Those reads were successfully used to (i) identify the species in culture, (ii) type the parasite’s group, and (iii) identify potential biomarkers of pharmacoresistance or virulence, reducing the computational cost and time in comparison to other strategies (i.e., assembly). Additionally, the longer read lengths than other sequencing alternatives (i.e., Illumina sequencing) permitted a reconstruction of the conserved region of the maxicircle sequences with as little as 3X coverage. Likewise, direct uncorrected nanopore reads provided sufficient coverage to identify chromosomal aneuploidies. These findings supported the presence of intrastrain mosaicism and interstrain diversity in our samples. Moreover, this methodology was able to determine the CNV status of 22 genes (divided in 10 loci) related to pharmacoresistance and virulence with shallow coverage (>5X). The analysis of additional strains with available phenotype data will be needed to validate LeishGenApp and the methodology presented here. Furthermore, the analyses of aneuploidy and CNV directly from clinical samples, coupled with in vitro drug resistance and pathogenicity tests, would help decipher the selected regions’ suitability as biomarkers for *L. infantum*.

## Figures and Tables

**Figure 1 microorganisms-10-02256-f001:**
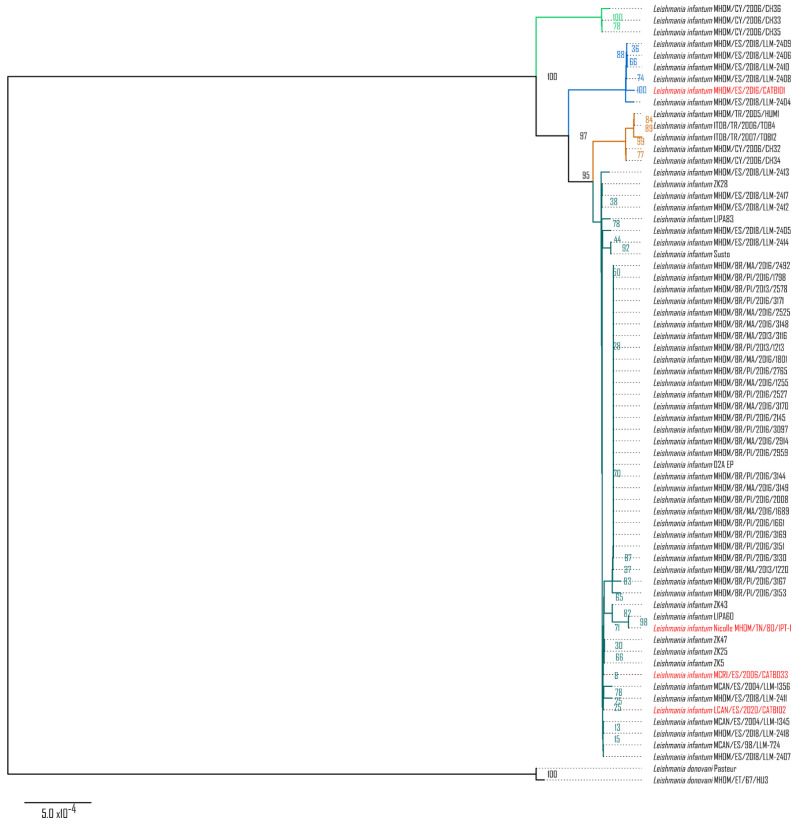
Maximum likelihood phylogenetic tree of the *L. donovani–L. infantum* group. *L. donovani–L. infantum* consensus phylogeny was constructed using the conserved region of the maxicircle, concatenating the coding sequences of 12S rRNA, 9S rRNA, ND8, ND9, MURF5, ND7, CO3, CYb, ATPase 6, ND2, G3, ND1, CO2, MURF2, CO1, G4, ND4, G5 (ND3), RPS12, and ND5. The maximum likelihood tree (log-likelihood −21,183.118) was modeled with the Tamura–Nei 1993 model with 1000 bootstraps. Taxa highlighted in red correspond to the novel maxicircle sequences of MHOM/TN/80/IPT-1, MHOM/ES/2016/CATB101, MCRI/ES/2006/CATB033, and LCAN/ES/2020/CATB102 samples. Subspecies phylogenetic clusters, types, or groups are highlighted in green, blue, yellow, and turquoise, corresponding to the same clusters identified in Solana et al., 2022 [12].

**Figure 2 microorganisms-10-02256-f002:**
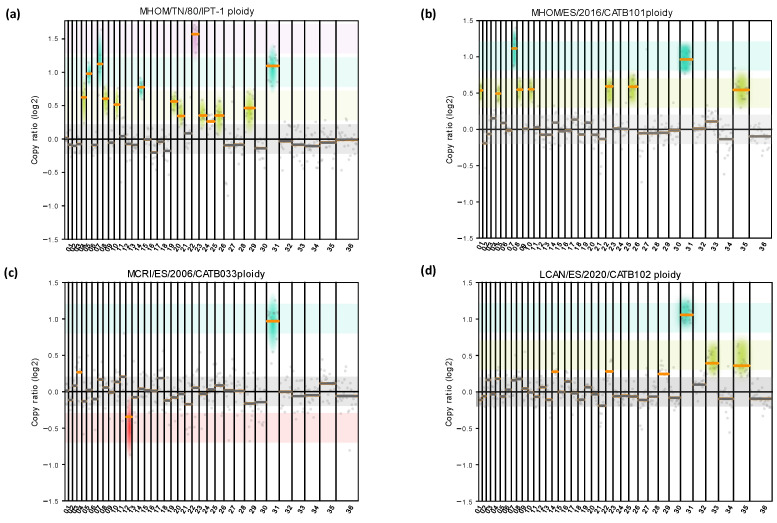
Detection and quantification of different aneuploidy patterns among *L. infantum* cultures. Aneuploidy analysis was carried out by adapting the sliding window size to 100 kbp (gray dots) for MHOM/TN/80/IPT-1 (**a**), MHOM/ES/2016/CATB101 (**b**), LCAN/ES/2020/CATB102 (**c**), and MCRI/ES/2006/CATB033 (**d**). Copy ratio represents the fold change of expected and observed chromosomal dotation. The median copy ratio is represented in orange if significantly different than 0 or in gray if not significant. Monosomy is highlighted with red shadowing, disomies with gray, trisomies with green, tetrasomies in turquoise, and hexasomies or larger in violet. No pentasomies were observed.

**Table 1 microorganisms-10-02256-t001:** **Sample origin and metadata used in this study.** Four cultured *L. infantum* samples MHOM/TN/80/IPT-1, MHOM/ES/2016/CATB101, LCAN/ES/2020/CATB102, and MCRI/ES/2006/CATB033 were used in this study. Columns represent the relation with their original host, type of sample, leishmaniosis presentation (visceral or VL, cutaneous or CL), organ of isolation, and year and geographical location.

Sample ID	Host	Type	Isolation	Year	Location
MHOM/TN/80/IPT-1	*Homo sapiens*	VL	Unknown	1980	Monastir, Tunisia
MHOM/ES/2016/CATB101	*Homo sapiens*	VL	Spleen	2016	Mallorca area, Spain
LCAN/ES/2020/CATB102	*Canis lupus familiaris*	CL, VL	Lymph node aspirate	2020	Zaragoza area, Spain
MCRI/ES/2006/CATB033	*Canis lupus familiaris*, sub. *Cricetus aureus*	CL	Exudate	2006	Spain

**Table 2 microorganisms-10-02256-t002:** **Summary of known biomarkers for drug resistance in *L. infantum***. Summary of locus, gene ID, function, and reported gene copy number variation of the nine genomic regions (22 loci) analyzed related to pathogenicity or drug resistance in *L. infantum*. Gene ID, Start, and End correspond to the *L. infantum* reference genome (*L. infantum* JPCM5 v2/2018) [18]. CNV: Expansion or deletion of gene copy number. R: Resistant; TF: Treatment failure. A reference is provided for studies showing threshold resistance effect by CNVs.

Biomarker	Gene ID	Start	End	Gene Name	Function	Gene CNVs	Resistance/Pathogenicity	Reference
Miltefosine sensitivity locus (MSL)	LinJ.31.2370	1,181,281	1,182,328	LinJ.31.2370	3′-nuclease	Deletion (CN −2)	Miltefosine TF	[27]
	LinJ.31.2380	1,184,204	1,185,341	LinJ.31.2380	3′-nuclease			
	LinJ.31.2390	1,185,826	1,188,553	LinJ.31.2390	Helicase-like protein			
	LinJ.31.2400	1,191,356	1,192,406	LinJ.31.2400	3-2-trans-enoyl-CoA isomerase			
Miltefosine transporter and associated genes	LinJ.13.1590	570,912	574,206	LdMT	Phospholipid transport	Deletion (CN −1, −2)	Miltefosine R	[28]
	LinJ.13.1600	576,108	577,572	Hypot. protein	Unknown			
	LinJ.32.1040	392,366	393,596	ldRos3	Vps23 core domain			
LACK antigen	LinJ.28.2940	1,070,377	1,071,316	LACK1	Antigenic protein	Expansion	Pathogenicity	[29]
	LinJ.28.2970	1,074,409	1,075,348	LACK2				
Paromomycin-resistant locus	LinJ.27.1940	942,538	944,020	D-LDH	D-lactate dehydrogenase	Expansion (CN +1)	Paromomycin (PMM) R	[30]
	LinJ.27.1950	946,545	947,751	B-CAT	Branched-chain amino acid aminotransferase			
MAPK1	LinJ.36.6760	2,564,560	2,565,637	LMPK	Mitogen-activated protein kinase	Conflicting evidence	Trivalent antimonials R	[31]
AQP1	LinJ.31.0030	8,742	9,687	AQP1	Aquaglyceroporin 1	Deletion (CN −1)	Trivalent antimonials R	[31]
H locus	LinJ.23.0280	86,372	86,942	YIP1	Unknown	Expansion (CN +1) MRPA and PTR1	Trivalent antimonials R	[28]
	LinJ.23.0290	88,619	93,329	MRPA	ABC-thiol transporter			
	LinJ.23.0300	94,265	95,522	LinJ.23.0300	Arginosuccinate synthase			
	LinJ.23.0310	101,314	102,181	PTR1	Pteridine reductase 1			
METK locus	LinJ.30.3550	1,283,752	1,284,865	Lorien protein		Deletion (CN −1)	Allopurinol R	[10]
	LinJ.30.3560	1,285,559	1,286,738	METK1	S-adenosylmethionine synthetase			
	LinJ.30.3570	1,288,872	1,289,985	Lorien protein				
	LinJ.30.3580	1,290,679	1,291,858	METK2	S-adenosylmethionine synthetase			

**Table 3 microorganisms-10-02256-t003:** Detection of CNV in 22 genes suitable as potential pharmacoresistance and pathogenicity biomarkers in *L infantum*. Variation in its local copy number for each sample, ranging between −1 and +7 gene copies compared to the diploid dotation (0). * Tetrasomy in chromosome 31 must be considered.

		MHOM/TN/80/IPT-1	MHOM/ES/2016/CATB101	MCRI/ES/2006/CATB033	LCAN/ES/2020/CATB102
**Miltefosine transporter and associated genes**		**CNV**	**CNV**	**CNV**	**CNV**
LinJ.13.1590	LdMT	0, +1	0, +1	−1	0
LinJ.13.1600	Hypot. Protein	+1	−1	0	0
LinJ.32.1040	ldRos3	−1	0	−1	0
**Miltefosine sensitivity locus (MSL)**		**CNV**	**CNV**	**CNV**	**CNV**
LinJ.31.2370 *	LinJ.31.2370	+4	+1, +2	+2	+2
LinJ.31.2380 *	LinJ.31.2380	+4	+1, +2	+2	+1, +2
LinJ.31.2390 *	LinJ.31.2390	+4, +5	+1, +2	+2	+1, +2
LinJ.31.2400 *	LinJ.31.2400	+3	+2	+2	+1, +2
**METK locus**		**CNV**	**CNV**	**CNV**	**CNV**
LinJ.30.3550	Lorien protein	0	+1, +2	+1	0
LinJ.30.3560	METK1	−1	+1	+1	0
LinJ.30.3570	Lorien protein	−1	+1	0, +1	−1
LinJ.30.3580	METK2	0	0, +1	0, +1	0
**H locus**		**CNV**	**CNV**	**CNV**	**CNV**
LinJ.23.0280	YIP1	+6, +7	+1	0, +1	+1, +2
LinJ.23.0290	MRPA	+5	+1, +2	0, +1	+1, +2
LinJ.23.0300	LinJ.23.0300	+5	+2	0	0
LinJ.23.0310	PTR1	+4	0, +1	−1	+1
**AQP1**		**CNV**	**CNV**	**CNV**	**CNV**
LinJ.31.0030 *	AQP1	+5	+1	+1, +2	+2, +3
**MAPK1**		**CNV**	**CNV**	**CNV**	**CNV**
LinJ.36.6760	LMPK	−1	0	−1	−1
**Amphotericin**		**CNV**	**CNV**	**CNV**	**CNV**
LinJ.2510	SMT	0	0, +1	0, +1	0
**Paramomycin-resistant locus**		**CNV**	**CNV**	**CNV**	**CNV**
LinJ.27.1940	D-LDH	0	0	0	0, +1
LinJ.27.1950	B-CAT	−1	0, +1	0	0, +1
**LACK antigen**		**CNV**	**CNV**	**CNV**	**CNV**
LinJ.28.2940	LACK1	0, +1	0, +1	0	0
LinJ.28.2970	LACK2	0, +1	0, +1	0	0, +1

## Data Availability

Raw sequencing data for this experiment can be accessed at the NCBI-NIH SRA deposit SRR21601459—SRR21601462, BioSamples SAMN30884654—SAMN30884657, under BioProject (SUB12055562).

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
