# Peer review of "Identification of Leishmania infantum Epidemiology, Drug Resistance and Pathogenicity Biomarkers with Nanopore Sequencing"

_microorganisms, 2022, doi:10.3390/microorganisms10112256_

Round 1
Reviewer 1 Report
The manuscript entitled "Identification of Leishmania infantum epidemiological, drug resistance and pathogenicity biomarkers with nanopore sequencing" is relevant and important to the field of study. I have a few concerns outlined below that would improve the manuscript for publication:
1. Results can be better described to improve readers' understanding.
2. The quality/resolution of figure 2 is low.
3. Discussion must be improved.
4. Conclusions should be more summarized; part of the conclusion should be in the discussion item.
Author Response
Dear reviewer,
Thank you very much for your willingness to read and comment on our preliminary work. Much appreciated.
- Results can be better described to improve readers' understanding.
- English grammar was revised in this section. L311-312 were modified to improve clarity.
- The quality/resolution of figure 2 is low.
- The resolution of Figure 2 was improved. It has been replaced already in-text.
- Discussion must be improved.
- Discussion was mildly modified to improve readability. Part of conclusions was included in this section.
- Conclusions should be more summarized; part of the conclusion should be in the discussion item.
- We edited part of the conclusions, avoiding possible redundancy with the discussion section.
The authors would like to appreciate the reviewer for this consideration of our article for publication.
Reviewer 2 Report
In the communication entitled "Identification of Leishmania infantum epidemiological, drug resistance and pathogenicity biomarkers with nanopore sequencing" the authors performed a study on twenty-two genes to generate a genetic drug resistance profile against miltefosine, allopurinol, antimonials, amphotericin and paromomycin. Leishmaniasis is a zoonotic parasitic disease caused in the Mediterranean area by the protozoan Leishmania infantum. The latter is responsible for canine leishmaniasis (LCan) and visceral and cutaneous leishmaniasis in humans. The parasite is transmitted to humans and dogs through the bite of vector insects belonging to the genus Phlebotomus spp., Commonly referred to as sand flies . The dog is the main domestic reservoir for the parasite. The emergence of drug-resistant strains of the parasite Leishmania infantum infecting dogs and humans place this issue in great public health concern as they represent a growing threat.
The manuscript provides information that is included in the scope of microbilology. The communication can be published in its current form in a high-level journal such as Microorganisms.
Author Response
Dear reviewer,
We'd like to extend our joy that you found our preliminary results interesting and worthy of publication in this journal. We hope it was found useful as well in your research.
Warm regards.